# The sub-annual calibration of hydrological models considering climatic intra-annual variations

Binru Zhao<sup>1, 2</sup>, Huichao Dai<sup>1</sup>, Dawei Han<sup>2</sup>, Guiwen Rong<sup>3</sup>

<sup>1</sup>College of Water Conservancy and Hydropower Engineering, Hohai University, Nanjing, 210098, China

<sup>5</sup> <sup>2</sup>Water and Environmental Management Research Centre, Department of Civil Engineering, University of Bristol, Bristol, BS81TR, UK

<sup>3</sup>College of Earth and Environment, Anhui University of Science and Technology, Huainan, 232001, China

Correspondence to: Binru Zhao (zbrhhu@gmail.com)

Abstract. Changing climate leads to change of temporal dynamics of hydrological systems by affecting the catchment conditions. Considering climatic variations when calibrating a hydrological model can improve model performance, which allows parameter sets to vary according to sub-periods with different climate conditions. This study has explored climatic intra-annual variations by using two classification approaches to recognize the sub-periods with similar climatic patterns, Calendar-Based Grouping (CBG) method and Fuzzy C-Means (FCM) algorithm. The model performances of the sub-annual calibration schemes based on these two approaches are compared using the conceptual model IHACRES. The effect of time

- scales on sub-annual calibration schemes was also studied. Results indicate that the sub-annual calibration scheme based on CBG method performs better than that based on Rainfall-dominated FCM algorithm, since the CBG method has a better performance in recognizing temperature pattern, and the main source of catchment change is from the change of vegetation, which is mainly affected by temperature in the study site. The optimal time scale is dependent on the sub-annual calibration scheme, with bimonthly for CBG method and Temperature-dominated FCM algorithm and seasonal for Rainfall-dominated
- FCM algorithm. Overall, when using sub-annual calibration schemes, the selection of the partitioning method and time scale is very important to model performances.

#### 1. Introduction

Understanding hydrological responses of a catchment is important to water resources managers. Hydrological models provide useful tools for this task, varying from simple statistical or conceptual models to complicated spatially-distributed or physically

based models. Although physically based hydrological models can better describe the real physical process, the conceptual hydrological models are widely used to address some management and research problems, because they rely on fewer parameters and have satisfactory performances.

Hydrological model parameters are generally estimated through calibration from the historically observed rainfall and streamflow data, and the model is then validated using the data outside of the calibration period. The parameters are assumed to be stationary for the historical period and these calibrated parameters are also assumed to be valid for the future period. However, this assumption is challenged due to the change of catchment conditions. The reliability of using stationary model parameters in a changing environment has been questioned in previous studies (e.g. Milly et al., 2008; Brigode et al., 2013), where the change of catchment is mainly indicated by the climate change since the climate change can directly and indirectly

affect the catchment conditions and the climate data can be obtained more easily.

Several attempts have been made to explore intra-annual variations of hydrological processes. Paik et al. (2005) proposed a

seasonal tank model where parameter sets were calibrated based on three 4-month seasons. The seasonal tank model gives much less calibration error than the non-seasonal model. Levesque et al. (2008) found that when winter and summer data were

- used separately to calibrate the SWAT model, the model performance was considerably improved over the case when only summer observations were provided for calibration. However, there is no real advantage when the model is calibrated based on winter observations compared with the traditional calibration using all available data. Luo et al. (2012) examined ten parameterization schemes at 12 catchments located in three different climatic zones in east Australia. The results show that it is worth calibrating the model with the use of data from each individual month for the purpose of seasonal streamflow
- forecasting.

On the other hand, there are some studies focusing on the inter-annual dynamics of hydrological behaviors. In such a scheme, different climatic conditions are taken into account. Klemeš (1986) initially considered the need to verify hydrological model parameters under different climate conditions. He proposed a differential split-sampling test in which two opposing climate

- periods were identified and the hydrological model was calibrated and validated by the contrast periods. Merz et al. (2011) applied the test to 273 catchments in Australia and found that the parameters representing snow and soil moisture processes showed high correlations to changing climatic conditions of the catchments. Consequently, the performance of the model is particularly affected if the calibration and validation periods differed substantially. The similar results were found by Brigode et al (2013), where the differential split-sampling test was used to group one wet, one medium, and two dry sub-periods on the
- basis of the Aridity Index, and each sub-period group was calibrated separately for two conceptual models, leaving one dry period for the validation. The results showed that the validation had the worst performance when the models were calibrated against the wet sub-period.
- Recently, calibrating model parameters based on a portion of the record with conditions similar to those of the future period to simulate is suggested. In this scheme, clustering methods were widely used to identify the periods with hydrological similarities. Choi and Beven (2007) classified the 30-day data sets into 15 clusters using Fuzzy C-Means algorithm. The TOPMODEL was calibrated and validated for each cluster in the GLUE framework. Although the evaluations showed satisfactory results at the global level, no parameter set was found satisfactory for all 15 clusters. De Vos et al. (2010) used the k-means clustering algorithm to partition the historical data into 12 clusters of hydrological similarity, allowing the model parameters to vary over
- clusters. They also improved the model structure by analyzing the patterns in the parameter sets of the various clusters. Toth (2009) classified hydro-meteorological conditions with a clustering method based on Self-Organising Maps (SOM). The results show that an adequate distinction of the hydro-meteorological conditions may considerably improve the rainfall-runoff modelling performance. On the other hand, when using the sub-annual calibration scheme, some researchers divided periods into several groups based on calendar. Luo et al. (2012) calibrated the hydrological model only using the data from the same
- 70 month and each month has one optimized parameter set. The results indicate this calibration scheme has better performance compared with other calibration schemes. Kim and Han (2016) compared the model performance under different sub-annual calibration schemes in terms of serial calibration scheme (SCS) and parallel calibration scheme (PCS). Parameter sets are estimated for each sub-period which is based on different time scales. They found that PCS performed slightly better than SCS.
- The above-mentioned studies discussed the reliability of hydrological models to simulate runoff under varying climate conditions and demonstrated that allowing model parameters to vary over time can improve the performance of hydrological models. In many cases, the historical period is divided into groups or clusters according to climate characteristics. When recognizing climate patterns that contribute to temporal dynamics of hydrological responses, there are generally two approaches, clustering methods and calendar-based method. Although sub-annual calibration schemes based on these two

approaches have significant advantages compared with the traditional calibration using all available data, the comparison of model performances between these two approaches has not been considered yet in the literature. Besides, the time period of the sub-period can range from month to year, which may have effects on model efficiencies, and there may be an optimal time scale for the sub-annual calibration scheme. When using clustering methods, a definition of the number of clusters is required before the clustering is performed. Therefore, how to define the number of clusters and its effect on the model performance also deserves exploration.

In this study, we only consider the climatic intra-annual variations through using two classifying approaches to recognize climatic patterns, Calendar-Based Grouping (CBG) method and Fuzzy C-Means (FCM) algorithm. Sub-annual calibration schemes based on these two approaches allow parameter sets to vary according to climatic patterns, which is based on four

time scales (biannual, seasonal, bimonthly and monthly). The conceptual hydrological model IHACRES is applied to one catchment in England to explore the above-mentioned problems.

## 2. Study Sites

The study site herein considered is the Thorverton catchment, located in the southwest of England. The Thorverton catchment is one of the Exe subcatchments with an area of around 606 km<sup>2</sup>. Figure 1 shows the overview of the Thorverton catchment.

- Reasons for selecting Thorverton catchment are (1) the catchment has a climate with warm, dry summers and cold, wet winters, showing great intra-annual variations in terms of rainfall and flow (Fig. 2); (2) the Thorverton catchment has a long history in meteorological and hydrological observations and there are available daily time series of rainfall, flow and temperature.
- The average daily rainfall data for the period 1890-2015 over the Thorverton catchment were obtained from NERC Environmental Information Data Centre (Tanguy et al., 2016). The daily time series of the observed flow data were available for the period from 1957 to 2014, which can be extracted from the National River Flow Archive (NRFA) provided by the Center for Ecology & Hydrology. The catchment average temperatures covering the period from 1960 to 2011 were calculated based on the daily temperature data at 50 km × 50 km grid cells, which were downloaded from the UKCP09 gridded observation data sets. The daily time series of rainfall, flow and temperature are all available for the period from 1960 to 2011.
- 05 This study used data from 1960 to 2000 to calibrate the hydrological model, and the rest for model validation.

#### 3. Methodology

#### 3.1 Hydrological model

The IHACRES model (Jakeman and Hornberger, 1993) is a conceptual rainfall-runoff model. The model has a satisfactory performance in simulating the catchment rainfall-runoff response as a function of total streamflow and has been widely applied to a range of catchments for climate impact studies due to its flexibility. (Jakeman et al., 1993; Letcher et al., 2001; Kim and

Lee, 2014; Kim et al., 2016).


The IHACRES model comprises two modules, in series: the non-linear loss module and the linear routing module. The nonlinear loss module transforms rainfall into effective rainfall. Effective rainfall is defined as the portion of rainfall that eventually

leaves the catchment as streamflow, which is then converted to the streamflow by the linear routing module. The linear routing module applies the well-known unit hydrograph theory, conceptualizing the catchment as a configuration of linear storages


acting in series and / or parallel. The structure of the IHACRES model is shown in Fig. 3. Model parameters are listed in Table1. More details about the model are described by Jakeman and Hornberger (1993).

### 3.2 Recognition of sub-periods with hydrological similarities

The traditional calendar-based sampling method is applied to divide historical observations into sub-periods at different time scales (biannual, seasonal, bimonthly and monthly). With regard to monthly time scale, for instance, the historical observations which include n years are sampled every one month, resulting in  $12 \times n$  sub-periods of the length of one month.

When recognizing sub-periods with hydrological similarities, two methods are used, Calendar-Based Grouping (CBG) method and Fuzzy C-Means (FCM) algorithm.

#### Calendar-Based Grouping (CBG) method

Calendar-Based Grouping (CBG) method assumes that the same calendar months or seasons among different years have similar climatic patterns, therefore, they can be classified into one group. In this scheme, sub-periods sampled from historical observations at monthly time scale are classified into 12 groups. And the numbers of groups for biannual, seasonal and bimonthly time scales are 2, 4 and 6, respectively.

#### Fuzzy C-Means (FCM) algorithm

Fuzzy C-Means (FCM) algorithm is an unsupervised clustering algorithm which was originally introduced by Bezdek (1981).

- 35 In clustering, objects with similar characteristics are classified into one cluster, and objects in different clusters are dissimilar in the same characteristics (Sbai, 2001; Pakhira et al., 2004). In this study, FCM algorithm was used to cluster sub-periods of historical observations by recognizing different climatic patterns based on five climate variables. These climate variables include four variables regarding rainfall and one regarding temperature. Periodic rainfall, maximum daily rainfall, rate of rainy days, and variance of rainfall form a representation of rainfall conditions, and the variable of periodic average temperature
- 40 examines the role of temperature in recognizing hydroclimatic patterns. These five climate variables are considered to be capable of describing climate characteristics of each sub-period.

When classifying sub-periods  $X = \{x_1, x_2, ..., x_n\}$  into k clusters denoted by fuzzy sets  $(F_j, j = 1, ..., k)$ , the algorithm is based on minimization of the following objective function

(1)

$$J_m = \sum_{i=1}^n \sum_{j=1}^k (\mu_{ij})^m \left\| x_i - c_j \right\|^2$$

where  $\mu_{ij}$  is the membership degree of  $x_i$  to the cluster  $F_j$  with  $\sum_j \mu_{ij} = 1$ .  $m \in [1, \infty)$  is a weight exponent controlling the degree of fuzzification.  $c_j$  is the cluster centroids of the fuzzy cluster  $F_j$ , and  $||x_i - c_j||$  is an Euclidean norm between  $x_i$  and  $c_j$ . In this study,  $x_i$  is the  $i^{th}$  climate variable of the targeted sub-period.

Fuzzy partitioning is conducted through an iterative optimization of the above-shown objective function, with the update of membership degree  $\mu_{ii}$  and the cluster centroids  $c_i$  until no further improvement in  $J_m$  is possible.

FCM algorithm needs initial definition of the number of clusters before the clustering is performed. An evaluation method is

required to determine the optimal number of clusters. In previous studies, the validity index ( $V_{\chi_B}$ ) proposed by Xie and Beni

(1991) is used to evaluate the validation of fuzzy c-partitions.

$$V_{XB} = \frac{\sum_{i=1}^{n} \sum_{j=1}^{n} \mu_{ij}^{2} \|x_{i} - c_{j}\|^{2}}{n\left(\min_{j \neq p} \|c_{j} - c_{p}\|^{2}\right)}$$

(2)

The optimal number of clusters  $k_{opt}$  is obtained by minimizing  $V_{XB}$  over k = 2,3,  $k_{max}$ . The value of  $k_{max}$  can be chosen according to pre-knowledge from the data set.

However, whether the model based on the optimal number of clusters defined by  $V_{XB}$  can really have the best simulation performance remains unknown. Therefore, models based on clusters with different numbers were run to assess the validation of  $V_{XB}$ .

#### 3.3 Sub-annual calibration scheme

For the traditional calibration scheme, the parameters remain stationary during the calibration period under the assumption that parameters are valid for the entire calibration period. For the sub-annual calibration scheme, parameter sets are allowed to vary according with different climate conditions. In this paper, it means that the parameter set is stationary for sub-periods in one group or cluster but differ among groups or clusters. Therefore, the model will be calibrated separately for each group or cluster. When calibrating parameters for one group or cluster, although the entire calibration periods are used to run the model, objective function only considers sub-periods in the specific group or cluster.



For this calibration scheme, the difference between the observed and simulated flow is minimized by maximizing the Nash-Sutcliffe Efficiency (NSE) (Nash and Sutcliffe, 1970) which is defined as:

$$NSE = 1 - \frac{\sum_{i=1}^{N} (Q_{obs,i} - Q_{sim,i})^2}{\sum_{i=1}^{N} (Q_{obs,i} - \overline{Q_{obs}})^2}$$
(3)

where,  $Q_{obs,i}$  and  $Q_{sim,i}$  are the simulated and observed runoff of the  $i^{ih}$  day, respectively.  $\overline{Q_{obs}}$  is the arithmetic mean of the observed runoff. N is the number of days in a specific group or cluster.

For the optimization algorithm, we initially used the Latin hypercube sampling method to generate various random parameter sets. The best combination of parameter values was chosen as the initial start of the nlminb function according to the NSE value of various parameter sets. The nlminb function was then applied to search the optimal parameter sets.

# 80 **3.4 Evaluation of sub-annual calibration scheme**

Model performance is assessed for both the calibration period and the validation period with NSE. The validation method is also based on sub-periods with hydroclimatic similarities, which is similar with calibration procedures. Firstly, sub-periods in the validation period are matched into the most similar cluster of all clusters in the calibration period and assigned the corresponding optimal parameter set. Secondly, each parameter set is applied separately to run the model for the whole

validation period. Thirdly, extract simulated runoff of specific sub-periods from each simulation to combine the final simulated

runoff.

#### 4. Results and discussion

### 4.1 Optimal number of clusters for FCM algorithm

The validity index  $V_{\chi B}$  evaluate the validation of fuzzy c-partitions from the respective of the compactness of the fuzzy

- 90 partition and the separation between clusters. When defining the optimal number of clusters,  $V_{XB}$  was calculated for each number of clusters which ranged from 2 to 15. The optimal number of clusters occurred where the value of  $V_{XB}$  is minimum. On the other hand, the optimal number of clusters was defined according to performances of each model which was run based on clusters with different numbers. The optimal number of clusters from two methods is shown in Table2.
- Figure 4 shows the variation of NSE for both the calibration period and validation period and the validity index  $V_{XB}$  with the number of cluster at monthly time scale, and other time scales have similar results. The NSE for calibration period generally increases with the number of clusters, which indicates the model calibrated to more clusters can better response the hydrological responses over the catchment, because the differences of hydrological responses among different climate patterns could be described through more runs of models which highlights specific climate pattern each time. There are no significant
- relationships between the value of NSE and  $V_{XB}$ , and the optimal number of clusters from  $V_{XB}$  and simulation for monthly time scale are 7 and 12 respectively.

Although the optimal number of clusters from two methods are different, the NSE for validation period of these two optimal numbers of clusters are very similar, 0.8081 and 0.8093 respectively. Therefore, the cluster validity index  $V_{XB}$  in identifying

the optimal number of clusters has a satisfactory performance and we used the cluster validity index  $V_{\chi B}$  to recognize the optimal number of clusters in this study.

#### 4.2 Partitioning of sub-periods with hydrological similarities

The difference between the Calendar-Based Grouping (CBG) method and Fuzzy C-Means (FCM) algorithm in partitioning sub-periods with hydrological similarities are shown in Fig. 5 and Fig. 6, respectively. Figure 5 describes the difference between the distribution of groups classified by the CBG method and clusters from the FCM algorithm for different time scales (biannually, seasonal, bimonthly and monthly) in the period 1990-1995. Clusters are numbered based on the value of periodical rainfall. We can found there are big differences in their distribution. Sub-periods in one group generally belonged to different clusters. Sub-periods in clusters with small rainfall statistics are more distributed in the period from March to August, while sub-periods in months from September to February are more classified into clusters with large rainfall statistics, which

indicates that the rainfall in the month of September, December, January and February is more than that of other months.

Figure 6 shows the distribution of climatic variables for each cluster and group from FCM algorithm and CBG method respectively, which is calculated from observations in the calibration period at monthly time scale. The width of boxes are related with the sample size of each cluster or group. For clusters, their sample sizes are variable and Cluster 2, Cluster 3,

Cluster 4 comprise more sub-periods compared with other clusters. For groups classified by the CBG method, they have the same number of sub-periods. Comparing the FCM algorithm with CBG method, for all climatic variables regarding rainfall,

the variations in clusters are smaller than those in groups and the median values of variables for different clusters range more greatly in comparison with those for different groups. However, the variable average temperature has opposite results. In terms of outliers in each cluster or group, FCM algorithm has a better performance because there are fewer outliers in clusters.

Overall, the FCM algorithm can better recognize different rainfall patterns and put together sub-periods with similar rainfall characteristics, while the CBG method has a better performance in recognizing the temperature pattern. The results also show that the rainfall in the month of September, December, January and February is more than that of other months.

## 4.3 Model efficiency of two sub-annual calibration schemes

The simulation performance of sub-annual calibration schemes based on two classifying approaches is compared in Fig. 7.
The traditional calibration method which calibrates the model using all historical observations was also performed with the purpose of comparison. The sub-annual calibration schemes based on hydroclimatic similarities show advantages in model performance over the traditional calibration scheme. Except for the biannual time scale, the simulation performance for both calibration period and validation period indicates that CBG method performed better than FCM algorithm. The reason for this might be that the catchment change in the study site is mainly affected by the temperature since CBG method has a better performance in recognizing sub-periods with similar temperature patterns (Fig. 6).

In order to prove this hypothesis, we have done an experiment where the FCM algorithm is performed only on the basis of temperature variables (maximum temperature, minimum temperature and average temperature for each sub-period). The distribution of climatic variables for each cluster shows big differences when indicators characterizing the climate differ (Fig.

8). The temperature-dominated FCM algorithm can better recognize sub-periods with similar temperature patterns with the optimal number of clusters 9.

Figure 9 compares the model efficiency of sub-annual calibration schemes based on three classifying approaches. The model performance is considerably improved using the Temperature-dominated FCM algorithm in comparison with Rainfalldominated FCM algorithm for both the calibration period and validation period. However, the difference of the model performance based on Temperature-dominated FCM algorithm and CBG method is not obvious. For the calibration period, the performance of the Temperature-dominated FCM algorithm is slightly better than CBG method except for bimonthly time scale, while for the validation period, except for monthly time scale, there is no improvement for the Temperature-dominated FCM algorithm compared to the CBG method.


Overall, the hypothesis that the catchment change in the study site is mainly affected by the temperature proves true. A possible interpretation of this phenomenon is that the main source of catchment change is from the change of vegetation, since the growth of vegetation (indicated by NDVI) is mainly impacted by temperature, which can be found from the positive correlation between temperature and NDVI in Fig. 10. The correlation coefficient 0.667 also indicates a positive correlation between

temperature and NDVI, which is calculated with the data for the period 2001-2011, while there is no significant correlation between rainfall and NDVI. Therefore, identifying the relevant climate factors to vegetation growth is of great importance for the clustering approach, since the vegetation change is the main source for the temporal change of the hydrological model parameters for this study catchment.

### 4.4 Evaluation of the optimal time scale

The optimal time scale for different sub-annual calibration schemes can be found from Fig. 9. For the calibration period, the optimal time scale is monthly for three sub-annual calibration schemes. However, for the validation period, the optimal time

scale is bimonthly for CBG method and Temperature-dominated FCM algorithm. Although Rainfall-dominated FCM algorithm has the optimal time scale of season, the performance of bimonthly time scale is very close to the optimal one.

Figure 11 shows the validation performance of the CBG-based calibration scheme for the period 2005-2008. The biannual time scale performed worst with NSE of 0.812, which resulted in underestimation for most periods, while the other three time scales all have a good performance, and the difference of these three time scales in model performance is very tiny. With regard to high flows, bimonthly time scale have a slight better result especially in November of 2006.

#### 5. Conclusion

- This study compared the hydrological model performance of different sub-annual calibration schemes, which take into account the intra-annual variations of climate. Two methods recognizing similar climatic pattern were applied to partition sub-periods with hydroclimatic similarities, Calendar-Based Grouping (CBG) method and Fuzzy C-Means (FCM) algorithm. The sub-annual calibration schemes based on hydroclimatic similarities exhibit advantages over the traditional calibration schemes is which assumes the catchment condition is stationary. However, the model performance of sub-annual calibration schemes is
- affected by the partitioning method and time scales.

The CBG method has a better performance in recognizing temperature pattern, while FCM algorithm performs better in recognizing rainfall pattern since partitioning indicators of the FCM algorithm are most related to rainfall conditions. It is found that the sub-annual calibration scheme based on the CBG method leads to a better model performance, which may

- 80 implies the catchment change in the study site is mainly affected by temperature conditions rather than rainfall conditions. This hypothesis is proved true through performing FCM algorithm only based on temperature variables, since the model performance is considerably improved using the Temperature-dominated FCM algorithm in comparison with Rainfalldominated FCM algorithm for both calibration period and validation period; however, the difference of the model performance for the Temperature-dominated FCM algorithm and CBG method is not obvious. A possible interpretation of this phenomenon
- is that the main source of catchment change is from the change of vegetation, since there is a positive correlation between temperature and NDVI over the study catchment. Therefore, identifying the relevant climate factors to vegetation growth is of great importance for the clustering approach. On the other hand, the cluster validity index  $V_{XB}$  is proved feasible in identifying the optimal number of clusters for the FCM algorithm. Additionally, the optimal time scale is dependent on the sub-annual calibration scheme, with bimonthly for the CBG method and Temperature-dominated FCM algorithm and seasonal for the
- 90 Rainfall-dominated FCM algorithm. Overall, when using sub-annual calibration schemes, the selection of partitioning method and time scale is very important to the model performance.

## Acknowledgement

This study was supported by Program for Changjiang Scholars and Innovative Research Team in University (IRT1233) and the Fundamental Research Funds for the Central Universities (2016B42014). We acknowledge the UK Met Office and Centre for Ecology & Hydrology for providing the data.

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

# 35

# Tables

# Table1: List of parameters in the IHACRES model

| Module      | Parameter           | Description                   | Minimum | Maximum |
|-------------|---------------------|-------------------------------|---------|---------|
| None-linear | С                   | Mass balance                  | 0       | 0.04    |
|             | $	au_w$             | Reference drying rate         | 0       | 50      |
| Linear      | f                   | Temperature modulation of     | 0       | 4       |
|             |                     | drying rate                   | 0       |         |
|             | l                   | Soil moisture index threshold | 0       | 50      |
|             | р