# Peer review of "The sub-annual calibration of hydrological models considering climatic intra-annual variations"

_Hydrology and Earth System Sciences, 2017_

## Referee Comment (RC1) · E. Toth (Referee) · 13 Oct 2017

**Referee Comments by Elena Toth**

**MS hess-2017-396**
**"The sub-annual calibration of hydrological models considering climatic intra-annual variations"**
**by Binru Zhao, Huichao Dai, Dawei Han, and Guiwen Ron**

The paper presents a comparison of procedures (based either on clustering or on calendar) for calibrating and validating a rainfall-runoff model with parameter sets that depend of the period of the year.

Since the analysis of the seasonal variability of the dominating hydrological processes is a crucial topic, and the importance of keeping such variability into account when calibrating a rainfall-runoff model is often neglected in both research and hydrological practice, the addressed theme is of broad interest for the HESS readers.

The application to only one case study (even if with adequately long time-series) is certainly a serious limitation of the work, as highlighted also by the Editor, Ralf Merz, in addition a number of clarifications on the work are needed.

The main concerns I have are:

1) The final objective of the procedure is not clear.
   The method is in fact not supposed to be used for choosing a different parameter set in real-time (like some examples in the cited literature, where the hydro-meteorological conditions PRECEDING the forecast instant are used for the classification/clustering and for the choice of the most adequate parametrization to be used for real-time forecasting), but it has to be used off-line, since the hydro-meteorological similarity is identified a posteriori in the clustering technique here applied. In fact, in order to identify the cluster to which a specific time instant belongs, the future rainfall (and temperature) values are needed in input.
   The sub-annual calibration scheme based on FCM is therefore applicable only a posteriori; such drawback, in addition to its complexity, is not justified by the results, since the calendar grouping performs equally well than the best FCM.
   Maybe the authors should elaborate more on the differences in the simulation results in comparison to the traditional approach, possibly in order to improve the model structure?
   The final aim of the study should be clearly stated in the introduction and a deeper interpretation of the results is needed in the result/conclusion sections.
2) The reasons for the choice of the variables used in the clustering technique are not clear: of course many other hydro-meteorological features may be needed to appropriately identify the peculiarity of each subperiod/ 'season'.

In addition, It is important to underline that an important problem in using the model with changing parameter sets is the fact that the model is a continuously-simulating conceptual one, that needs all the previous simulation values (depending on the specific parameter set) to update the state variables. This also implies that when switching from one subperiod to the following one (e.g. at the end of the month and beginning of the new one) there may be some discontinuities in the simulated streamflow, due to the change. Such aspects are one of the main issues in the use of time-varying parameters in rainfall-runoff modelling and it's not very clear in the presentation.

p. 1, 32-35: please be careful with the use and meaning of the terms 'stationarity' and 'climate change':

see, Lins' note (2012) on the WMO website: http://www.whycos.org/chy14/download/file.php?id=13

in particular, in this case, the study does not address climate change, but interannual variations, so I don't think such digression (especially being a very complex and debated issue) is needed.

Section 3.2: a flowchart or a diagram explaining the splitting and use of the different time periods would be very useful to understand the proposed approaches.

p.4: ll 30-31: explain how the months are merged: the 6-months periods are only the Jan to June one and the July to December one, or other 6 consecutive months periods have been analysed?

p. 4, ll 36-37: specify that the clustering technique was applied for all the time-scales (1 month, 2 months, 4 months and 6 months)

p. 4, ll 37-41: please add more information on the selection of the input variables: which other variables have been considered, how you have chosen such five ones, etc; (see point 2) above)

p. 5, ll 60-62: add that the description of the steps for identifying Kopt is reported in Section 4.1.

p.5, ll 68-69: as said above, explain that the problem in using the model with changing parameters is the fact that the model is a continuously-simulating conceptual one, that needs all the previous simulation values (depending on the specific parameter set) to update the state variables: for this reason the model has to be run for the entire observation period and not only for the analysed sub-period.

p. 5, ll. 77-79: more information on the optimization algorithm are needed and in particular either add the the definition and meaning of of 'nlminb', or remove such detail.

p. 5, l.84- 85: explain better how dealing with discontinuities in the simulated streamflow values when going from one period to the following one (see comments above).

All section 4.1 must be thoroughly revised and reworded since it's very confusing and the utility of using the cluster validity index is far from demonstrated (in the only information referring to it, Fig. 4, the values of $V_{XB}$ seem to fluctuate randomly):

- from ll. 90-93 and ll. 100-06 it is not clear how, eventually, the optimal number of clusters is identified, considering both the simulation results and the validity index;
- l. 92: with 'according' you mean 'also considering'?
- ll.96-97: comment also on the results for the validation period.
- Overall, the text does not report the final chosen value for Kopt at monthly time-scale, and most importantly, nor the text nor Table 2 report the final number of clusters chosen for each of the other time-scales: in table 2, both possible values of Kopt are shown for each time-scale. And the paper does not provide any information on the reasons for the choice of the number of clusters for all the other time-scales (2-, 4-, 6—months periods), since Figure 4 (in addition to reporting difficult to interpret results) refers only to the monthly time-scale (even if this is not stated in the caption).

p. 6, l. 11: why Fig. 5 refers only to the years 1990-1995? The calibration period is 1990-2000.

Fig. 5 does not provide any information on the relation between clusters and seasons: you should find a way to show this information and to analyse it deeper.

p. 6, l. 12: with 'sub-periods in one group' you mean the 'sub-periods in the same calendar position'?

p.7, l. 25-26: this result is hardly surprising: the first FCM is based on four over five input variables that depend only on rainfall, whereas temperature has a clear annual cycle, well-reproduced by a calendar method. And in fact Fig. 6 is not very useful, since all the plots show the same pattern in the rainfall variables. Probably more/different climatic variables would provide more insights in the hydrological behavior of the catchment during the year (see point 2)

Section 4.3 (ll 36-49): adding a second FCM classification procedure based on different climatic variables as a sort of 'second thought' experiment in the results section makes the overall work difficult to follow: please introduce also such FCM algorithm earlier, in section 3, together with the other two techniques, and not here when discussing the results.

End of section 4.3: please add considerations also on the possible effect of snow (guided by temperature) in the study basin.

p. 8, l. 65: why Fig. 11 shows the simulation for the period 2005-2008? The validation period is 2001 to 2011.

p. 8, ll.76-82: also this paragraph suffers in clarity from the 'late' addition of a second FCM scheme: rephrase referring to both FCM algorithms as developed at the same time and with the same 'dignity'.

p. 8, ll. 87-88: actually, given the confusion and lack of information in section 4.1, this conclusion (utility of cluster validity index for choosing Kopt) is not supported by what is presented in the current version of the manuscript.

p. 8, ll. 89-90 (and section 4.4): please elaborate more on such result (bi-monthly sub-periods as the best performing partition), trying to explain it, if possible.

---

## Referee Comment (RC2) · Anonymous Referee #2 · 14 Oct 2017

Review of the article:

**The sub-annual calibration of hydrological models considering climatic intra-annual variations**

By **Binru Zhao, Huichao Dai, Dawei Han and Guiwen Rong**
Submitted for possible publication in *Hydrology and Earth System Sciences.*

**1 GENERAL COMMENTS**

The manuscript presents the results of different calibration procedures that are based on climatic similarities between sub-periods and on one rainfall-runoff model, methods applied on a unique catchment in UK. The issue of the rainfall-runoff model parameter dependence to the climatic period considered for the calibration is very interesting, especially in the context of the quantification of the climate change impacts on hydrology. Thus, the paper subject is highly relevant and the tested methodology is interesting and original, but the paper is lacking significant information about the applied methodology, the studied catchment and is lacking elements on how this methodology could be applied in an operational context. Moreover, the consideration of only one catchment is a strong limitation of this paper and is not enough discussed in the conclusion. Some of the paper figures are useless; the other ones are poorly presented in the paper and in their caption. These comments are detailed in the first part of this review and specific comments are given in the second part.

**1.1 Studied catchment**

Considering only one catchment for such study is a strong limitation for the generalization of the obtained results. Why not considering other catchments and applying the same methodology on an ensemble of different catchments?
The paper lacks some justification on the choice of this particular catchment regarding the objectives of the study. What are the particularities of this catchment in terms of hydro-climatic variability (both inter and intra-annual)? Moreover, information on the quality of the studied times series is lacking. The potential temporal variability of the measurement quality is highly important in such studies. For example, the poor hydrometric quality of the flow time series on several particular years could significantly affect the performance of the rainfall-runoff model calibration on this sub-period and thus misleading the result interpretation.
Finally, the presentation of the catchment regime and of the temporal variability of the hydro-climatic series (flow, temperature and precipitation) is lacking.

**1.2 Calibration methodology**

The presentation of the developed methodology is lacking some important information and the applied methodology presents some limitations that need be discussed.
Considering only one calibration and evaluation criterion in a study based on only one catchment is somehow disappointing. Why only looking at the Nash and Sutcliffe (1970) Efficiency (NSE) criterion? I think that considering the Kling and Gupta Efficiency score (KGE, Gupta *et al.*, 2009) and analyzing its different sub-criterions will be interesting for studying the benefits of the different calibration procedures in terms of flow mean bias, variance bias and temporal correlation…
The choice of the calibration and validation periods is important in this type of study. Why the selection of this particular periods for calibration (1960-2000) and validation (2001-2011)? Why only using 10 years for validation and why not considering different validation periods?
Is it not clear to me why you did not choose an index considering both precipitation and temperature variables for grouping periods, such as the aridity index, cited in the introduction section and used by Brigode *et al.* (2013)?
In addition, I think that performing a calibration on a "randomly grouping" for each time steps would be an interesting reference to compare with climatic grouping.

In the subsection 3.4 (line 182 to 184), you stated that the "sub-periods in the validation period are matched into the most similar cluster of all clusters in the calibration period". This point needs to be discussed. What about potential differences between clusters of the calibration period and validation period? What about potential new clusters? This should be addresses in the results section by comparing the characteristics of the calibration and validation sub-periods.

Finally, it is unclear how the model parameters are obtained for each calibration process. I think that you should explain how you perform a continuous rainfall-runoff simulation over a given period and how you calibrate the model only over several timesteps and sub-period.

**1.3   Seasonal bias of the model?**

I think that an analyze of the seasonal performances of the model should be added before applying the different calibration strategies, as an analyze of the performance on the different sub-periods considered. For example, the calculation of NSE for each season and each month would be interesting. Thus, potential seasonal biases in the rainfall-runoff model calibration could be identified and discussed.

**1.4   Use of "only" one hydrological model**

Could you please discuss the fact that you only considered one rainfall-runoff model in this study? What would be the conclusion if you applied the same calibration methodology with one other hydrological rainfall-runoff?

**1.5   Operational use of this methodology?**

Could you please discuss the potential uses of your developed methodology in applied studies? How this method could be applied for the quantification of the climate change impacts of catchment hydrology? For each catchments?

**2   SPECIFIC COMMENTS**

- **Line 27:** could you detail what you mean by "satisfactory performances"?
- **Line 31:** could you detail what you mean by "stationary"? Such word has to be clearly defined in this context of climate change.
- **Line 32:** could you detail what you mean by "catchment conditions": climatic, land use, hydrological conditions?
- **Line 32 to 35:** this sentence is very unclear. I think that you should be more precise on what you mean by "change of catchment", "climate change" and "catchment conditions"…
- **Line 36 to 38:** please give more details on what is a "calibration error" and if validation performances have been quantified in this study, and on which catchments the methodology has been applied.
- **Line 38 to 42:** again, on how many catchments, where (and thus in which climate) this test has been conducted? How many years of calibration were available? Are these results obtained in calibration or in validation on an independent sub-period?
- **Line 44:** "worth" compared to what? The report of the conclusion of this paper is unclear although it seems particularly interesting considering the aim of the submitted paper.
- **Line 48:** what is "different climatic" conditions?
- **Line 51:** Merz *et al.* (2011) worked on catchments in Austria and not in Australia.
- **Line 53:** could you clarify that the difference between calibration and validation periods are in terms of climate?
- **Line 55**: could you clarify what is, in this context, the aridity index and how it is calculated?
- **Line 55:** what is a "sub-period group" in this context?
- **Line 56:** again, could you explain what you called "performances" here? In terms of what score?

- **Line 61:** what is a "30-day data sets" in this context?
- **Line 64:** could you clarify what is an "hydrological similarity"?
- **Line 65:** do you refers to Toth and Brath (2007) instead of Toth (2009)?
- **Line 71:** could you clarify what are the difference between the "serial" and the "parallel" calibrations in this context?
- **Figure 1**: this figure needs to be strongly improved, with the addition of:
  - a general map of the UK,
  - a scale bar,
  - the elevation of the catchment,
  - the position of the rivers and of the gauging station.
- **Line 94:** the Figure 2 needs to be more deeply presented, with explanation on the period considered and on the obtained results, for example.
- **Line 110:** could you define what the word "flexibility" means in this context? Also, the "," after flexibility needs to be deleted.
- **Figure 3:** this figure seems to be useless. I think that a complete diagram of the rainfall-runoff model with the different parameters would be more useful.
- **Table 1:** please add parameter units.
- **Line 119:** please consider to change the title of this subsection into "recognition of… with climatic similarities" since you choose your sub-periods based only on climatic variables. I think that this change has to be made all over the paper.
- **Line 124:** please consider changing "hydrological" into "climatic".
- **Line 129 to 131:** please consider to merge these two sentences and rephrase them, since they are unclear to me.
- **Line 138:** could you clarify what is the "periodic rainfall" variable?
- **Line 154:** please cite the "previous studies" you mentioned.
- **Line 160 to 161:** thus, why considering this validity index (cf. section 1.2 of this review) ?
- **Line 164:** again, I think that you should clarify and define first in the introduction what you mean by "stationary" or you should avoid this word.
- **Line 164 to line 169:** this paragraph lacks some clear explanation on how model parameters are obtained (cf. section 1.2 of this review).
- **Line 176 to 179:** this paragraph is unclear: why using Latin Hypercube Sampling? What is the *nlminb* function?
- **Line 182:** no, the similarities of sub-periods are only "climatic" and not "hydroclimatic" in your approach.
- **Line 185:** please rewrite this unclear sentence.
- **Line 194:** could you give some explanation on the obtained results presented in the Table 2?
- **Figure 4:** please correct the figure legend by writing "calibration". You should explicitly state in the figure caption that these results are obtained with the monthly time scale.
- **Line 196:** I think that the results obtained with the other timesteps are interesting and may be somehow presented in the paper. Please consider to add these results in the paper.
- **Line 196 to 199:** please rewrite this unclear sentence.
- **Line 207:** please change "hydrological similarities" into "climatic similarities".
- **Line 209:** please change "hydrological similarities" into "climatic similarities".
- **Figure 5:** I do not understand how the figure 5 presents the difference between two classifications. For me it only shows the results of one classification. Moreover, why only presenting this 5-year period? This has to be addressed in the paper. Finally, why the number of groups is different considering different time steps?
- **Line 209 to 215:** this paragraph is unclear. It seems to me that the authors are in the end analyzing the rainfall regime through the calibrations results, while a basic analysis of the observed regimes (cf. section 1.1 of this review) would a priori give the same information.
- **Figure 6:** please indicate the quantiles used for the construction of the boxplots. What are the points outside of the boxplots? Please give the scale of the box widths, which is proportional

to the size of the group. Please also state explicitly in the caption legend that you are presenting the results for the monthly time step only.

- **Line 223 to 224:** could you define what is a outlier in this context and why you consider that better performance are obtained for classification with "fewer outliers in clusters" ?
- **Line 225 to 227:** It seems to me that the CBG is, by construction, better able to capture the flow seasonal pattern since it exists a clear seasonal pattern for the temperature of the studied catchment, while there is no clear seasonal pattern for precipitation. The climatic regimes of the catchment needs to be plotted and presented before (cf. sections 1.1 and 1.3 of this review). Please consider this observation for the analysis of the Figure 7. Finally, why not showing the same figure for the other time steps, for which the results could be less obvious?
- **Line 231:** change "hydroclimatic" to "climatic".
- **Line 233 to 235:** this sentence needs to be clarified and strongly improved in terms of explanation quality (cf. section 1.1 of this review). Is there any indication of climatic change on the studied catchment or is there "only" a seasonal bias in the rainfall-runoff model performances?
- **Line 243 to 249:** are you sure that this "new" classification method obtained with the "temperature-dominated FCM algorithm" is not the same classification that the calendar-based one?
- **Line 251 to 258:** this paragraph needs to be improved or deleted. Please state what is NDVI, where and how you define this index. Why did you analyze the correlation over the 2001-2011 period? Why only a sub-period is plotted on the Figure 10?
- **Figure 11**: why only this sub-period (2005-2008) is plotted and why only the CBG calibration is considered? This figure is useless in this form, since it is difficult to compare the calibration strategies.
- **Line 273**: change "hydroclimatic" into climatic.
- **Line 274:** define or delete the "stationary" word.

**3   REFERENCES**

Brigode, P., Oudin, L., Perrin, C., 2013. Hydrological model parameter instability: A source of additional uncertainty in estimating the hydrological impacts of climate change? Journal of Hydrology 476, 410–425. doi:10.1016/j.jhydrol.2012.11.012

Gupta, H.V., Kling, H., Yilmaz, K.K., Martinez, G.F., 2009. Decomposition of the mean squared error and NSE performance criteria: Implications for improving hydrological modelling. Journal of Hydrology 377, 80–91. doi:10.1016/j.jhydrol.2009.08.003

Merz, R., Parajka, J., Blöschl, G., 2011. Time stability of catchment model parameters: Implications for climate impact analyses. Water Resour. Res. 47, W02531. doi:10.1029/2010WR009505

Nash, J.E., Sutcliffe, J.V., 1970. River flow forecasting through conceptual models part I—A discussion of principles. Journal of hydrology 10, 282–290.

---

## Author Comment (AC1) · 13 Nov 2017

Dear Dr.Toth,

We appreciate the helpful and constructive comments and our responses to the comments are as follows:

The paper presents a comparison of procedures (based either on clustering or on calendar) for calibrating and validating a rainfall-runoff model with parameter sets that depend of the period of the year.

Since the analysis of the seasonal variability of the dominating hydrological processes is a crucial topic, and the importance of keeping such variability into account when calibrating a rainfall-runoff model is often neglected in both research and hydrological practice, the addressed theme is of broad interest for the HESS readers.

The application to only one case study (even if with adequately long time-series) is certainly a serious limitation of the work, as highlighted also by the Editor, Ralf Merz, in addition a number of clarifications on the work are needed.

**Reply:** Agreed. We will apply the proposed methodology to additional two catchments (with different geophysical characteristics) to explore its generalisation capabilities.

The main concerns I have are:

(1) The final objective of the procedure is not clear.

The method is in fact not supposed to be used for choosing a different parameter set in real-time (like some examples in the cited literature, where the hydro-meteorological conditions PRECEDING the forecast instant are used for the classification/clustering and for the choice of the most adequate parametrization to be used for real-time forecasting), but it has to be used off-line, since the hydro-meteorological similarity is identified a posteriori in the clustering technique here applied. In fact, in order to identify the cluster to which a specific time instant belongs, the future rainfall (and temperature) values are needed in input.

The sub-annual calibration scheme based on FCM is therefore applicable only a posteriori; such drawback, in addition to its complexity, is not justified by the results, since the calendar grouping performs equally well than the best FCM.

**Reply:** We believe that the proposed methodology can be used in both off-line and real-time conditions. During a real-time situation, both temperature and precipitation for the event to be modelled could be obtained through real-time observation and weather nowcasting. However, we apologise for this misunderstanding due to the unclear description about the operational use of the proposed methodology, and will amend this in the revised manuscript.

Maybe the authors should elaborate more on the differences in the simulation results in comparison to the traditional approach, possibly in order to improve the model structure?

**Reply:** The simulation results of the proposed methodology have been compared with that of the traditional approach (Figure 7), which shows great improvements. This study aims to use more appropriate calibration schemes to compensate the deficiency of the model structure and invariant model parameters when considering seasonal variations of the catchment. How to improve the model structure may be explored in the future research.

The final aim of the study should be clearly stated in the introduction and a deeper interpretation of the results is needed in the result/conclusion sections.

**Reply:** The final operational aim of the study is to build a more appropriate hydrological model for water resource management (e.g., river flow extension by rainfall runoff modelling) or real-time flood forecasting (via data assimilation). Because hydrometeorological features/indices are considered, this study may also be useful for future rainfall-runoff modelling under climate change. We will add this in the revised manuscript as well as a deeper interpretation of the results.

(2) The reasons for the choice of the variables used in the clustering technique are not clear: of course many other hydro-meteorological features may be needed to appropriately identify the peculiarity of each subperiod/ 'season'.

**Reply:** Agreed. We will explore a wider range of hydrometeorological features and justify their use in the revised manuscript.

(3) It is important to underline that an important problem in using the model with changing parameter sets is the fact that the model is a continuously-simulating conceptual one, that needs all the previous simulation values (depending on the specific parameter set) to update the state variables. This also implies that when switching from one subperiod to the following one (e.g. at the end of the month and beginning of the new one) there may be some discontinuities in the simulated streamflow, due to the change. Such aspects are one of the main issues in the use of time-varying parameters in rainfall-runoff modelling and it's not very clear in the presentation.

**Reply:** Many thanks for highlighting this important discontinuity issue in changing model parameters. This is caused by the parallel running of models with different parameter sets. If the models are run in series, there will be no such discontinuity problem because the subsequent flow is mainly derived from the antecedent flow and part of the new effective rainfall. We will provide a better clarification in the revised manuscript.

**Specific comments:**

p. 1, 32-35: please be careful with the use and meaning of the terms 'stationarity' and 'climate change': see, Lins' note (2012) on the WMO website: http://www.whycos.org/chy14/download/file.php?id=13 in particular, in this case, the study does not address climate change, but interannual variations, so I don't think such digression (especially being a very complex and debated issue) is needed.

**Reply:** Many thanks for highlighting this important issue. We will pay more attention to this problem to avoid the digression in the revised manuscript.

Section 3.2: a flowchart or a diagram explaining the splitting and use of the different time periods would be very useful to understand the proposed approaches.

**Reply:** Agreed. We will add this diagram.

p.4: ll 30-31: explain how the months are merged: the 6-months periods are only the Jan to June one and the July to December one, or other 6 consecutive months periods have been analysed?

**Reply:** The 6-month periods are only the Jan to June one and the July to December one. This will be further explained in the revised manuscript.

p. 4, ll 36-37: specify that the clustering technique was applied for all the time-scales (1 month, 2 months, 4 months and 6 months).

**Reply:** Agreed. We will specify it.

p. 4, ll 37-41: please add more information on the selection of the input variables: which other variables have been considered, how you have chosen such five ones, etc; (see point 2) above)

**Reply:** Agreed. We will explore a wider range of hydrometeorological features and justify their use in the revised manuscript.

p. 5, ll 60-62: add that the description of the steps for identifying Kopt is reported in Section 4.1.

**Reply:** Agreed. We will add this description.

p.5, ll 68-69: as said above, explain that the problem in using the model with changing parameters is the fact that the model is a continuously-simulating conceptual one, that needs all the previous simulation values (depending on the specific parameter set) to update the state variables: for this reason the model has to be run for the entire observation period and not only for the analysed sub-period.

**Reply:** Agreed. The current calibration process is based on the parallel model run, and we will replace it with the series run for better modelling continuity. This will be clarified in the revised manuscript.

p. 5, ll. 77-79: more information on the optimization algorithm are needed and in particular either add the definition and meaning of 'nlminb', or remove such detail.

**Reply:** Agreed. We will provide more information on the optimization algorithm.

p. 5, l.84- 85: explain better how dealing with discontinuities in the simulated streamflow values when going from one period to the following one (see comments above).

**Reply:** This issue is caused by the parallel running of models with different parameter sets. If the models are run in series, there will be no such discontinuity problem because the subsequent flow is mainly derived from the antecedent flow and part of the new effective rainfall. We will replace the parallel run with the series run for better modelling continuity and provide a better clarification in the revised manuscript.

All section 4.1 must be thoroughly revised and reworded since it's very confusing and the utility of using the cluster validity index is far from demonstrated (in the only information referring to it, Fig. 4, the values of $V_{XB}$ seem to fluctuate randomly):

- from ll. 90-93 and ll. 100-06 it is not clear how, eventually, the optimal number of clusters is identified, considering both the simulation results and the validity index;

**Reply:** The purpose of this section is to verify the utility of the index $V_{XB}$ in identifying the optimal number of clusters for FCM algorithm through comparing with the value of NSE under different numbers of clusters. We apologise for this misunderstanding due to the unclear description and will amend this in the revised manuscript.

- l. 92: with 'according' you mean 'also considering'?

**Reply:** Yes, 'according' here means 'also considering'.

- ll.96-97: comment also on the results for the validation period.

**Reply:** Agreed. We will comment on the results for the validation period.

- Overall, the text does not report the final chosen value for Kopt at monthly time-scale, and most importantly, nor the text nor Table 2 report the final number of clusters chosen for each of the other time-scales: in table 2, both possible values of Kopt are shown for each time-scale. And the paper does not provide any information on the reasons for the choice of the number of clusters for all the other time-scales (2-, 4-, 6—months periods), since Figure 4 (in addition to reporting difficult to interpret results) refers only to the monthly time-scale (even if this is not stated in the caption).

**Reply:** Agreed. Although line 204 to 207 (Therefore, the cluster validity index $V_{XB}$ in identifying the optimal number of clusters has a satisfactory performance and we used the cluster validity index $V_{XB}$ to recognize the optimal number of clusters in this study) reports the cluster validity index $V_{XB}$ is used to recognize the optimal number of clusters, we did not report the final chosen value of Kopt for different time scales. We will provide more information on the choice of the number of clusters for all the time scales and revised this section. We will also consider other better ways to define the number of clusters for all time scales, since the results of the cluster validity index are not good enough in this study.

p. 6, l. 11: why Fig. 5 refers only to the years 1990-1995? The calibration period is 1990-2000.

**Reply:** The calibration period (1960-2000) is too long to exhibit in the figure, so we only choose the period 1990-1995 to clarify the difference between the distribution of groups classified by the CBG method and clusters from the FCM algorithm for different time scales.

Fig. 5 does not provide any information on the relation between clusters and seasons: you should find a way to show this information and to analyse it deeper.

**Reply:** Agreed. We will improve Figure 5 to better present the information on the relation between clusters and seasons and analyse it deeper.

p. 6, l. 12: with 'sub-periods in one group' you mean the 'sub-periods in the same calendar position'?

**Reply:** Yes. We will state it more clearly.

p.7, l. 25-26: this result is hardly surprising: the first FCM is based on four over five input variables that depend only on rainfall, whereas temperature has a clear annual cycle, well-reproduced by a calendar method. And in fact Fig. 6 is not very useful, since all the plots show the same pattern in the rainfall variables. Probably more/different climatic variables would provide more insights in the hydrological behavior of the catchment during the year (see point 2)

**Reply:** Agreed. We will explore a wider range of hydrometeorological features and justify their use in the revised manuscript.

Section 4.3 (ll 36-49): adding a second FCM classification procedure based on different climatic variables as a sort of 'second thought' experiment in the results section makes the overall work difficult to follow: please introduce also such FCM algorithm earlier, in section 3, together with the other two techniques, and not here when discussing the results.

**Reply:** Agreed. We will reorganize this paper, in which all the calibration schemes are introduced and discussed together.

End of section 4.3: please add considerations also on the possible effect of snow (guided by temperature) in the study basin.

**Reply:** Snow is a fairly rare occurrence in the studied catchment, so we did not consider the effect of snow.

p. 8, l. 65: why Fig. 11 shows the simulation for the period 2005-2008? The validation period is 2001 to 2011.

**Reply:** The data of the validation period is too many to show in the figure, so we only choose the period 2005-2008 to represent the validation period. However, this figure is not very clear to describe the results, and an improved figure will be provided in the revised manuscript.

p. 8, ll.76-82: also this paragraph suffers in clarity from the 'late' addition of a second FCM scheme:

rephrase referring to both FCM algorithms as developed at the same time and with the same 'dignity'.

**Reply:** Agreed. We will reorganize the paper, in which all the calibration schemes are introduced and discussed together.

p. 8, ll. 87-88: actually, given the confusion and lack of information in section 4.1, this conclusion (utility of cluster validity index for choosing Kopt) is not supported by what is presented in the current version of the manuscript.

**Reply:** We will further clarity the utility of cluster validity index for choosing Kopt in the revised manuscript.

p. 8, ll. 89-90 (and section 4.4): please elaborate more on such result (bi-monthly sub-periods as the best performing partition), trying to explain it, if possible.

**Reply:** Agreed. We will add more elaboration and explanation on this result.

We hope our responses to the comments are satisfactory and look forward to more suggestions.

Best regards,

Binru Zhao, the corresponding author

---

## Author Comment (AC2) · 13 Nov 2017

We appreciate the helpful and constructive comments and our responses to the comments are as follows:

**1 GENERAL COMMENTS**

The manuscript presents the results of different calibration procedures that are based on climatic similarities between sub-periods and on one rainfall-runoff model, methods applied on a unique catchment in UK. The issue of the rainfall-runoff model parameter dependence to the climatic period considered for the calibration is very interesting, especially in the context of the quantification of the climate change impacts on hydrology. Thus, the paper subject is highly relevant and the tested methodology is interesting and original, but the paper is lacking significant information about the applied methodology, the studied catchment and is lacking elements on how this methodology could be applied in an operational context. Moreover, the consideration of only one catchment is a strong limitation of this paper and is not enough discussed in the conclusion. Some of the paper figures are useless; the other ones are poorly presented in the paper and in their caption. These comments are detailed in the first part of this review and specific comments are given in the second part.

**1.1 Studied catchment**

Considering only one catchment for such study is a strong limitation for the generalization of the obtained results. Why not considering other catchments and applying the same methodology on an ensemble of different catchments?

**Reply:** Agreed. We will apply the proposed methodology to additional two catchments (with different geophysical characteristics) to explore its generalization capabilities.

The paper lacks some justification on the choice of this particular catchment regarding the objectives of the study. What are the particularities of this catchment in terms of hydro-climatic variability (both inter and intra-annual)? Moreover, information on the quality of the studied times series is lacking. The potential temporal variability of the measurement quality is highly important in such studies. For example, the poor hydrometric quality of the flow time series on several particular years could significantly affect the performance of the rainfall-runoff model calibration on this sub-period and thus misleading the result interpretation.

**Reply:** The studied catchment has evident intra-annual variations in terms of rainfall, flow and temperature, which have been shown in Figure 2 and Figure 6. As to the inter-annual variations, Figure 11 has illustrated the significant variations between different years (2005-2007). Because of such large inter-annual variations, the hydrometeorological feature cluster method as proposed in the paper should adapt to the catchment change better than the calendar based method. As to the data quality, we have carefully checked all the hydrometeorological data for possible outliers, missing data, etc. We will add all these in the revise manuscript.

Finally, the presentation of the catchment regime and of the temporal variability of the hydro-climatic series (flow, temperature and precipitation) is lacking.

**Reply:** Figure 1 has presented the catchment map and will be improved with the addition of more information. Currently, Figure 2 shows the temporal variability of flow and precipitation. As to the temporal variability of temperature, Figure 6 presents a clear seasonal pattern for the temperature. We will add the time series plots of flow, temperature and precipitation to better illustrate their temporal variations.

**1.2 Calibration methodology**

The presentation of the developed methodology is lacking some important information and the applied methodology presents some limitations that need be discussed.

Considering only one calibration and evaluation criterion in a study based on only one catchment is somehow disappointing. Why only looking at the Nash and Sutcliffe (1970) Efficiency (NSE) criterion? I think that considering the Kling and Gupta Efficiency score (KGE, Gupta et al., 2009) and analyzing its different sub-criterions will be interesting for studying the benefits of the different calibration procedures in terms of flow mean bias, variance bias and temporal correlation.

**Reply:** As mentioned above, more catchments will be added. Additional criteria will also be added.

The choice of the calibration and validation periods is important in this type of study. Why the selection of this particular periods for calibration (1960-2000) and validation (2001-2011)? Why only using 10 years for validation and why not considering different validation periods?

**Reply**: Agreed. The current division of the calibration and validation data can be improved by 3-fold cross validation so that all the data will have a chance to be used in calibration and validation.

Is it not clear to me why you did not choose an index considering both precipitation and temperature variables for grouping periods, such as the aridity index, cited in the introduction section and used by Brigode et al. (2013)?

In addition, I think that performing a calibration on a "randomly grouping" for each time steps would be an interesting reference to compare with climatic grouping.

**Reply:** Agreed. More hydrometeorological features/indices will be explored in the revised manuscript. We will consider the suggested 'random grouping' approach.

In the subsection 3.4 (line 182 to 184), you stated that the "sub-periods in the validation period are matched into the most similar cluster of all clusters in the calibration period". This point needs to be discussed. What about potential differences between clusters of the calibration period and validation period? What about potential new clusters? This should be addresses in the results section by comparing the characteristics of the calibration and validation sub-periods.

**Reply:** Agreed. This is a common problem with any data-based methods when the validation data is very different to the calibration data. However, even in such situations, the proposed method should still be better than the conventional invariant model approach because the nearest catchment model parameters to the validation period can be selected. We will add this clarification in the revised manuscript.

Finally, it is unclear how the model parameters are obtained for each calibration process. I think that you should explain how you perform a continuous rainfall-runoff simulation over a given period and how you calibrate the model only over several timesteps and sub-period.

**Reply:** The current calibration process is based on the parallel model run, and we will replace it with the series run for better modelling continuity. This will be clarified in the revised manuscript.

**1.3 Seasonal bias of the model?**

I think that an analyze of the seasonal performances of the model should be added before applying the different calibration strategies, as an analyze of the performance on the different sub-periods considered. For example, the calculation of NSE for each season and each month would be interesting. Thus, potential seasonal biases in the rainfall-runoff model calibration could be identified and discussed.

**Reply:** Agreed. We will add this analysis.

**1.4 Use of "only" one hydrological model**

Could you please discuss the fact that you only considered one rainfall-runoff model in this study? What would be the conclusion if you applied the same calibration methodology with one other hydrological rainfall-runoff?

**Reply:** Due to the constraints of time and resources, it is not feasible to explore multiple hydrological models in this study. IHACRES is a well known model widely used in hydrology, so the results would be of interest to the community. We hope this paper will stimulate more studies using the proposed methodology with more hydrological models.

**1.5 Operational use of this methodology?**

Could you please discuss the potential uses of your developed methodology in applied studies? How this method could be applied for the quantification of the climate change impacts of catchment hydrology? For each catchment?

**Reply:** The final operational aim of the study is to build a more appropriate hydrological model for water resource management (e.g., river flow extension by rainfall runoff modelling) or real-time flood forecasting (via data assimilation). Because hydrometeorological features/indices are considered, this study may also be useful for future rainfall-runoff modelling under climate change. We will add this in the revised manuscript.

**2 SPECIFIC COMMENTS**

Line 27: could you detail what you mean by "satisfactory performances"?

**Reply:** "Satisfactory performances" means although the conceptual hydrological models are not as good as the physically based hydrological models in modelling runoff, it is feasible to use them to address some management and research problems.

Line 31: could you detail what you mean by "stationary"? Such word has to be clearly defined in this context of climate change.

**Reply:** There was a wrong use of the term "stationary" in the original paper. We will replace it with "invariant".

Line 32: could you detail what you mean by "catchment conditions": climatic, land use, hydrological conditions?

**Reply:** "Catchment conditions" means land use or cover here, and it will be clarified in the revised manuscript.

Line 32 to 35: this sentence is very unclear. I think that you should be more precise on what you mean by "change of catchment", "climate change" and "catchment conditions".

**Reply:** Agreed. We will pay more attention to the use of these terms in the revised manuscript.

Line 36 to 38: please give more details on what is a "calibration error" and if validation performances have been quantified in this study, and on which catchments the methodology has been applied.

**Reply:** We will provide more details on this study we cited.

Line 38 to 42: again, on how many catchments, where (and thus in which climate) this test has been conducted? How many years of calibration were available? Are these results obtained in calibration or in validation on an independent sub-period?

**Reply:** We will provide more details on this study we cited.

Line 44: "worth" compared to what? The report of the conclusion of this paper is unclear although it seems particularly interesting considering the aim of the submitted paper.

**Reply:** Agreed. We will state the conclusion of this paper more clearly.

Line 48: what is "different climatic" conditions?

**Reply:** Different climatic conditions are identified by different values of the climate parameters of interest here. This will be clarified in the revised manuscript.

Line 51: Merz et al. (2011) worked on catchments in Austria and not in Australia.

**Reply:** Agreed. We will change "Australia" to "Austria".

Line 53: could you clarify that the difference between calibration and validation periods are in terms of climate?

**Reply:** We will add this clarification.

Line 55: could you clarify what is, in this context, the aridity index and how it is calculated?

**Reply:** We will add this clarification.

Line 55: what is a "sub-period group" in this context?

**Reply:** We will state it more clearly.

Line 56: again, could you explain what you called "performances" here? In terms of what score?

**Reply:** The performance here is evaluated by the NSEsq values. This will be clarified in the revised manuscript.

Line 61: what is a "30-day data sets" in this context?

**Reply:** "30-day data sets" means 30-day-period data sets sampled from hydrological time series with the moving window method. We will state it more clearly.

Line 64: could you clarify what is an "hydrological similarity"?

**Reply:** Hydrological similarity here is identified based on three variables: precipitation, the 10-day moving average of the precipitation and the GR4J-simulated soil moisture. We will add this clarification.

Line 65: do you refers to Toth and Brath (2007) instead of Toth (2009)?

**Reply:** We refer to Toth and Brath (2007). However, there is a mistake in the references. We will amend it.

Line 71: could you clarify what are the difference between the "serial" and the "parallel" calibrations in this context?

**Reply:** This clarification will be added in the revised manuscript.

Figure 1: this figure needs to be strongly improved, with the addition of:

- a general map of the UK,

- a scale bar,

- the elevation of the catchment,

- the position of the rivers and of the gauging station.

**Reply:** Agreed. We will improve this figure with the addition of these details.

Line 94: Figure 2 needs to be more deeply presented, with explanation on the period considered and on the obtained results, for example.

**Reply:** Agreed. We will add these details.

Line 110: could you define what the word "flexibility" means in this context? Also, the "." after flexibility needs to be deleted.

**Reply:** "Flexibility" means that one can define new soil moisture accounting models, new routing models, new calibration methods, new objective functions, and new evaluation statistics, while retaining as much of the default framework as is useful. And as the package code is available under an open source licence, one always has the freedom to adapt it as required. The "." after flexibility will be deleted.

Figure 3: this figure seems to be useless. I think that a complete diagram of the rainfall-runoff model with the different parameters would be more useful.

**Reply:** Agreed. We will improve this figure to better describe the IHACRES model.

Table 1: please add parameter units.

**Reply:** Agreed. We will add parameter units in this table.

Line 119: please consider to change the title of this subsection into "recognition of… with climatic similarities" since you choose your sub-periods based only on climatic variables. I think that this change has to be made all over the paper.

**Reply:** Agreed. We will change "hydrological similarities" to "climatic similarities".

Line 124: please consider changing "hydrological" into "climatic".

**Reply:** Agreed. We will change "hydrological" to "climatic".

Line 129 to 131: please consider to merge these two sentences and rephrase them, since they are unclear to me.

**Reply:** Agreed. These two sentences will be rephrased.

Line 138: could you clarify what is the "periodic rainfall" variable?

**Reply:** "Periodic rainfall" here means the accumulated rainfall during the specific period. This will be clarified in the revised manuscript.

Line 154: please cite the "previous studies" you mentioned.

**Reply:** Agreed. Previous studies will be cited.

Line 160 to 161: thus, why considering this validity index (cf. section 1.2 of this review) ?

**Reply:** The reason for using this validity index is that it proves valid in identify the optimal number of clusters for the FCM algorithm. However, the results of this index is not good enough in this study, so we will consider other better ways to identify the optimal number of clusters for FCM algorithm.

Line 164: again, I think that you should clarify and define first in the introduction what you mean by "stationary" or you should avoid this word.

**Reply:** Agreed. There is a wrong use of the term "stationary". We will replace it with "invariant".

Line 164 to line 169: this paragraph lacks some clear explanation on how model parameters are obtained (cf. section 1.2 of this review).

**Reply:** The current calibration process is based on the parallel model run, and we will replace it with the series run for better modelling continuity. This will be clarified in the revised manuscript.

Line 176 to 179: this paragraph is unclear: why using Latin Hypercube Sampling? What is the nlminb function?

**Reply:** Agreed. We will provide more information on the optimization algorithm.

Line 182: no, the similarities of sub-periods are only "climatic" and not "hydroclimatic" in your approach.

**Reply:** Agreed. We will replace "hydroclimatic" with "climatic".

Line 185: please rewrite this unclear sentence.

**Reply:** Agreed. We will rephrase this sentence.

Line 194: could you give some explanation on the obtained results presented in the Table 2?

**Reply:** We will give more explanation on the obtained results presented in the Table 2.

Figure 4: please correct the figure legend by writing "calibration". You should explicitly state in the figure caption that these results are obtained with the monthly time scale.

**Reply:** Apologize for this spelling mistake. We will correct it and state that these results are obtained with the monthly time scale.

Line 196: I think that the results obtained with the other timesteps are interesting and may be somehow presented in the paper. Please consider to add these results in the paper.

**Reply:** Agreed. We will add these results.

Line 196 to 199: please rewrite this unclear sentence.

**Reply:** We will rewrite this sentence.

Line 207: please change "hydrological similarities" into "climatic similarities".

**Reply:** We will change "hydrological similarities" into "climatic similarities".

Line 209: please change "hydrological similarities" into "climatic similarities".

**Reply:** We will change "hydrological similarities" into "climatic similarities".

Figure 5: I do not understand how the figure 5 presents the difference between two classifications. For me it only shows the results of one classification. Moreover, why only presenting this 5-year period? This has to be addressed in the paper. Finally, why the number of groups is different considering different time steps?

**Reply:** We will improve Figure 5 to better present the difference between two classifications. The calibration period (1960-2000) is too long to exhibit in the figure, so we only choose the period 1990-1995 to clarify the difference between the distribution of two classifications. The number of clusters are defined using the cluster validity index $V_{XB}$, and different time scales have different results, reported in Table 2. We will add these clarifications in the revised manuscript.

Line 209 to 215: this paragraph is unclear. It seems to me that the authors are in the end analyzing the rainfall regime through the calibrations results, while a basic analysis of the observed regimes (cf. section 1.1 of this review) would a priori give the same information.

**Reply:** This paragraph aims to state the difference of the distribution of groups classified by two methods through analyzing the rainfall regime. We apologize for the unclear description and will amend it.

Figure 6: please indicate the quantiles used for the construction of the boxplots. What are the points outside of the boxplots? Please give the scale of the box widths, which is proportional to the size of the group. Please also state explicitly in the caption legend that you are presenting the results for the monthly time step only.

**Reply:** Agreed. We will add these details in the revised manuscript.

Line 223 to 224: could you define what is a outlier in this context and why you consider that better performance are obtained for classification with "fewer outliers in clusters"?

**Reply:** Outliers represent sub-periods whose climate patterns differ a lot from others in each cluster or group. Fewer outliers in clusters indicate that the FCM algorithm could better recognize the similar climate patterns with less differences among the climate patterns of sub-periods.

Line 225 to 227: It seems to me that the CBG is, by construction, better able to capture the flow seasonal pattern since it exists a clear seasonal pattern for the temperature of the studied catchment, while there is no clear seasonal pattern for precipitation. The climatic regimes of the catchment needs to be plotted and presented before (cf. sections 1.1 and 1.3 of this review). Please consider this observation for the analysis of the Figure 6. Finally, why not showing the same figure for the other time steps, for which the results could be less obvious?

**Reply:** We will add the time series plots of flow, temperature and precipitation and use them to analyze the Figure 6. The results for other time scales is similar to that of the monthly scale, so we did not show the same figures for them.

Line 231: change "hydroclimatic" to "climatic".

**Reply:** Agreed. We will change "hydroclimatic" to "climatic".

Line 233 to 235: this sentence needs to be clarified and strongly improved in terms of explanation quality (cf. section 1.1 of this review). Is there any indication of climatic change on the studied catchment or is there "only" a seasonal bias in the rainfall-runoff model performances?

**Reply:** Agreed. We will improve this sentence. As to the climate change, the temperature rise during the study period is neglectable compared with its natural variations between seasons. We will further clarify it in the revised manuscript.

Line 243 to 249: are you sure that this "new" classification method obtained with the "temperature-dominated FCM algorithm" is not the same classification that the calendar-based one?

**Reply:** The temperature-dominated FCM algorithm can better adapt to the temperature inter-annual variations than the calendar based one.

Line 251 to 258: this paragraph needs to be improved or deleted. Please state what is NDVI, where and how you define this index. Why did you analyze the correlation over the 2001-2011 period? Why only a sub-period is plotted on the Figure 10?

**Reply:** Agreed. We will add this clarification. As to the period we use, the NDVI data is not available

for all periods and the results of a sub-period could be generalized to other periods, so we only choose the period 2001-2011 whose data quality is high to analyze the correlation. The period plotted on the Figure 10 is chosen randomly for the layout of the figures. We will pay more attention to this issue in the revised manuscript.

Figure 11: why only this sub-period (2005-2008) is plotted and why only the CBG calibration is considered? This figure is useless in this form, since it is difficult to compare the calibration strategies.

**Reply:** The data for all the validation period is too many to show in a figure, so we only choose a sub-period to analyze the results. The reason for only considering the CBG method is that it performs better than other two approaches in this study. We will add the analysis for other approaches in the revised manuscript. Figure 11 will also be plotted in a better way to show the differences between the simulated flow and observed flow for different calibration schemes

Line 273: change "hydroclimatic" into climatic.

**Reply:** Agreed. We will change "hydroclimatic" into "climatic".

Line 274: define or delete the "stationary" word.

**Reply:** Agreed. We will replace "stationary" with "invariant".

We hope our responses to the comments are satisfactory and look forward to more suggestions.

Best regards,

Binru Zhao, the corresponding author

---

## Author Comment (AC3) · 13 Nov 2017

Dear Dr.Merz,

Many thanks for your constructive comments which will be very helpful in improving the manuscript. Our responses to the comments are as follows:

Before potential publication in HESS I think it is necessary (next to the reviewer's comments) to address the following issues.

• Please discuss how your results can be generalized. You just analyse one catchment. Would be the ranking of the methods different for another catchment?

**Reply:** Agreed. We will apply the proposed methodology to additional two catchments (with different geophysical characteristics) to explore its generalisation capabilities.

• You can not see much in Figure 11. By showing all the time period there is hardly any differences between simulated and observed and hardly any differences between the various calibration schemes. Please focus on different time periods, so that one can really see differences between simulated and observed (and discus the reason for the differences) or try to increase the information content in another way. (or skip the figure).

**Reply:** Agreed. Figure 11 will be plotted in a better way to show the differences between the simulated flow and observed flow for different calibration schemes. The reason for the differences will also be discussed in the revised manuscript.

• Is the selected catchment a good one for demonstrating time variant model parameters? Is there any change in the climate? I think it is not shown in the paper.

**Reply:** Figure 7 has shown that the selected catchment has varied model parameters depending on the time. As to the climate change, the temperature rise during the study period is neglectable compared with its natural variations between seasons. We will further clarify all these in the revised manuscript.

We hope our responses to the comments are satisfactory and look forward to more suggestions.

Best regards,

Binru Zhao, the corresponding author